# Extracellular vesicles isolated from the plasma of COVID-19 and sepsis patients: characterisation and association with clinical outcomes

Jaques Franco Novaes de Carvalho[1], Paula Meneghetti[2], Gabriela Rodrigues Barbosa[1], Marina Malheiros Araújo Silvestrini[3], Sidneia Sousa Santos[1], Flávio Geraldo Freitas[4], Daniela Boschetti[4], Nancy Cristina Junqueira Bellei[1], Andréa Teixeira de Carvalho[3], Ana Claudia Torrecilhas[2]/+, Reinaldo Salomao[1]/+

[1]Universidade Federal de São Paulo, Infectologia da Escola Paulista de Medicina, São Paulo, SP, Brasil
[2]Universidade Federal de São Paulo, Instituto de Ciências Ambientais, Químicas e Farmacêuticas, Departamento de Ciências Farmacêuticas, Laboratório de Imunologia Celular e Bioquímica de Fungos e Protozoários, Diadema, SP, Brasil
[3]Fundação Oswaldo Cruz-Fiocruz, Instituto René Rachou, Belo Horizonte, MG, Brasil
[4]Hospital Sepaco, São Paulo, SP, Brasil

**BACKGROUND** Extracellular vesicles (EVs) are involved in the pathogenesis of severe acute respiratory syndrome Coronavirus 2 (SARS-CoV-2) infection.

**OBJECTIVES** We analysed the concentration, size, cellular origin, and capacity for carrying viral components in plasma samples from patients with Coronavirus disease 2019 (COVID-19) and sepsis.

**METHODS** Plasma samples from COVID-19 patients admitted to the intensive care unit (ICU) with sepsis (N = 42) and healthy individuals (N = 19) were analysed. EVs were characterised by size and concentration using nanoparticle tracking analysis (NTA), polymerase chain reaction (RT-qPCR) for SARS-CoV-2 components, and flow cytometry for immunophenotyping. EVs were marked with phosphatidylserine and tetraspanins. Cellular origin markers were used for neutrophils, endothelial cells, T lymphocytes and platelets. Cryo-EM was used to assess EV size and integrity.

**FINDINGS** NTA showed an increased concentration of microparticles in patients. RT-qPCR analysis of EVs detected the virus in 14 samples, two of which were consistent with the Gamma variant. EVs predominantly derived from T cells and platelets and demonstrated an increased expression of CD81 in individuals who died. Cryo-EM revealed EVs with an average size of 200 nm.

**MAIN CONCLUSIONS** Our findings suggest that patients' EVs likely harboured viral components, suggesting their potential role as carriers of SARS-CoV-2. In addition, EVs from deceased patients demonstrated elevated levels of CD81 expression.

Key words: extracellular vesicles - SARS-CoV-2 - COVID-19 - sepsis

Since 2003, coronaviruses have been the cause of serious and fatal diseases, including large-scale epidemic outbreaks. They were first named as severe acute respiratory syndrome Coronavirus (SARS-CoV), which started the epidemic in China in 2003. Soon after, in 2012, an epidemic was caused by Middle East Respiratory Syndrome Coronavirus (MERS-CoV). On December 31, 2019, the World Health Organisation (WHO) was alerted to cases of pneumonia in the city of Wuhan, Hubei province, in China. The disease, characterised as the Coronavirus disease 2019 (COVID-19), was caused by a coronavirus called SARS-CoV-2, which spread worldwide in 2020 and caused a global catastrophe, with millions of lives lost, and the collapse of public health systems. It evolved to become the biggest pandemic of the last 100 years.[1,2]

Extracellular vesicles (EVs) are spherical particles that vary in size from 30 nm to 2 μm, are covered by a bilipid layer formed from the plasma membrane and contain the cytosol of the secretory cell. They are also secreted by pathogens and plants.[3,4] EVs are present in various biological fluids, such as blood, saliva, milk, urine, ascites, cerebrospinal fluid, bronchoalveolar fluid and cerebrospinal fluid. They are related to several physiological processes, including intercellular communication. They contain nucleic acids, such as mRNA, microRNA and non-coding RNAs, as well as glycoconjugate proteins, lipids and metabolites.[5] According to the International Society of Extracellular Vesicles (ISEV), which established particle classification parameters based on identification, size, morphological assessment, and origin, EVs are classified into three subtypes: exosomes, microvesicles and apoptotic bodies.[3,6] Exosomes are smaller and are divided into large exosomes (Exo-L), from 90 to 120 nm, and small exosomes (Exo-S), from 60 to 80 nm.[7] Microvesicles or multivesicular bodies (MVB), measuring between 100

Financial support: FAPESP [Grants 2017/21052-0 and 2020/05110-2 (to RS), Grant 2020/07870-4 (to ACT)].
+ Corresponding authors: ana.torrecilhas@unifesp.br | https://orcid.org/0000-0001-5724-2199/ rsalomao@unifesp.br | https://orcid.org/0000-0003-1149-4598

**Handling editor:** Adeilton Alves Brandão | https://orcid.org/0000-0001-5877-607X

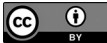

and 1000 nm, are fragments originating from the plasma membrane found on the cell surface.[8] Apoptotic bodies, measuring between 500 and 2000 nm, are found on the cell surface of cell membrane evaginations of apoptotic cells.[9] The composition of EVs is generally enriched with the presence of some trafficking components, such as Tsg101, Alix, phosphatidylserine, and those derived from transmembrane proteins called tetraspanins. Tetraspanins (CD9, CD63 and CD81) are generic markers of exosomes and microvesicles. EVs contain a wide variety of cell-specific markers from which the origin of the vesicles can be identified.[10]

It has been demonstrated that exosomes from virus-infected cells containing mRNA and miRNA proteins can act in virus propagation and modulation of the immune response. The human immunodeficiency virus (HIV), exosomes from HIV-infected cells, which release the viral protein Nef, can trigger the induction of apoptosis of CD4+ T lymphocytes, contributing to the pathogenesis of the virus.[11] In fact, EVs play an important role in the pathogenesis of sepsis, acute respiratory distress syndrome (ARDS), chronic obstructive pulmonary disease (COPD), pulmonary hypertension (PH) and other diseases.[12] To date, the role of the SARS-CoV-2 virus in EV trafficking is unclear but experiments with SARS-CoV-1 *in vitro* have shown EVs from infected alveolar cells that contain virions.[13] Electron microscopy images of lung tissue from a deceased patient with COVID-19 showed SARS-CoV-2 viral particles in double-membrane EVs.[14] Kwon and colleagues demonstrated that EVs isolated from lung epithelial cells contained the SARS-CoV-2 viral RNAs of non-structural proteins (nsp1, nsp12) and structural proteins (E and N).[15]

Patients with COVID-19 present a wide spectrum of clinical manifestations, including asymptomatic, mild, moderate, severe and critical illness.[16] Patients infected with SARS-CoV2 that progresses to critical COVID-19 illness unequivocally present sepsis, yet clinical courses and pathogenesis might differ in several aspects.[17]

In this study, we propose to examine the existence and source of EVs in the plasma of critically ill COVID-19 patients with sepsis, whether they function as carriers of viral components, and how they are associated with clinical outcomes.

## SUBJECTS AND METHODS

*Casuistic* - Forty-two (N = 42) individuals with symptoms of COVID-19 that fulfilled the criteria of sepsis[18] were enrolled in a prospective cohort upon admission to the intensive care unit (ICU) in a tertiary hospital, Hospital Sepaco in São Paulo, Brazil, from March to August 2021. All patients presented positive polymerase chain reaction (RT-qPCR) tests for COVID-19, and none of them had positive bacterial blood cultures at admission. Nineteen healthy donors (N = 19), matched for age and gender, were included as controls. Patients and volunteers were enrolled after they (or their legal guardians) signed informed consent forms. The demographic and clinical data of the patients were recorded and collected on the Research Electronic Data Capture (REDCap) platform. The study was approved by the local ethical committee (CAAE: 58045422.5.1001.5505).

*Blood samples* - 40 mL of peripheral blood was collected from patients and healthy donors in tubes containing EDTA (K2) (BD Biosciences, USA). After collection, part of the sample was centrifuged for 15 min at 2500 x g at 4ºC to obtain plasma, which was frozen at -80ºC and preserved until required.

*Isolation of total EVs from plasma of COVID-19 patients and healthy donors* - Around 2 mL of plasma was thawed, and the samples were centrifuged at 40,000x g for 30 min at 4ºC. After centrifugation, the supernatant was removed and subjected to ultracentrifugation at 100,000 x g for 16 h at 4ºC in an Optima L-100 XP ultracentrifuge with an SW 41Ti rotor (Beckman Coulter, USA). The pellet obtained was resuspended in 200 µL of phosphate-buffered saline (PBS) (sterile 0.2 µm membrane) and stored at -20ºC.

*Characterisation of total circulating EVs or microparticles by nanoparticle tracking analysis (NTA)* - To determine the average dimensions and concentrations of particles, samples of total circulating EVs or nanoparticles from COVID-19 patients and healthy controls were diluted 100-fold in filtered PBS and introduced into the NanoSight LM10 apparatus (Malvern Panalytical, UK), which was equipped with an sCMOS camera, a 532 nm green laser, with automatic camera height adjustment and manual calibration of the focus and detection threshold as required. All measurements were documented in triplicate for 60 s, each sample utilising the NTA program (version 2.3, Malvern Panalytical, UK).

*Characterisation of total circulating EVs and phenotyping by flow cytometry* - Characterisation and cellular origin of total circulating EVs: 20 samples from infected donors and 10 samples from healthy donors were selected from patients with the highest concentrations of particles/ml measured in NTA. Samples were incubated with phosphatidylserine [Annexin V-BV605 (BD - Becton Dickinson, USA)] and monoclonal antibodies conjugated with tetraspanins [CD9 (PE-CF594), CD81 (BV711) (BD - Becton Dickinson, USA)]. To enable the observation of the EV population, the forward scatter (FSC) and side scatter (SSC) parameters were adjusted based on magnetic bead calibrators of varying diameters of 100 nm, 160 nm, 200 nm, 240 nm, 300 nm, 500 nm and 900 nm by the Gigamix size standard (BioCytex, a Stago group company, FRA) in fluorescence in fluorescein isothiocyante (FITC). To verify the cellular origin of EVs by immunophenotyping, the samples were incubated with Annexin V-BV605 to detect EVs, and specific antibodies for neutrophils, T lymphocytes, platelets and endothelium, as shown in Table I.

When adding the markers according to their previously tested dilutions for EVs, 50 µL of EVs from each sample and 250 µL of 1X Annexin V-BV605 binding buffer were added to each tube and incubated for 30 min in the dark at room temperature. Soon after, the samples were run on the Cytoflex S flow cytometer (Beckman Coulter, USA) located in the Laboratory at the René Rachou Institute Flow Cytometry Platform - FIOCRUZ - in Minas

Gerais. EV samples were acquired with a minimum of 1,000,000 events. Data was analysed using the Flowjo v10.9.0 software (Becton Dickinson, USA). The analysis strategy for characterising and identifying the cellular origin of EVs was defined by calibrating the "gate" with Gigamix reference beads (FITC) and the sample team, then selecting the gate using two-dimensional graphs of Annexin V-BV605 versus granularity (SSC - by violet laser) [(Supplementary data (Fig. 1)]. The markers of interest were quantified as percentage values of EVs, comparing the initial marking of Annexin V-BV605 carried out in the absence of markers to that carried out in the presence of markers delimiting positivity[19] [Supplementary data (Fig. 2)]. One-dimensional histograms of gigamix beads and specific markers were overlaid to verify the size profile of EVs from 100 to 900 nm for each EV marker and cellular origin [(Supplementary data (Fig. 3)].

*Verification of the presence of SARS-CoV-2 viral components by RT-qPCR* - Total RNA was extracted from EV isolates using the QIAamp Viral RNA Mini Kit (Qiagen, Germany, Cat. #52904). RNA extraction was performed according to the manufacturer's instructions. The presence of SARS-CoV-2 viral components was verified in the isolated EVs using RT-qPCR Kits of different specificities. The first was the RT-qPCR kit (GeneFinder Kit, OSANG Healthcare, KOR), which amplifies RNA-dependent RNA polymerase (RdRp), Envelope (E) and Nucleocapsid (N) genes. The second was the AgPath one-step RT-qPCR Kit (in house) [Thermo Scientific, USA], which targets the E and N genes. Both kits include a reverse-transcription step that generates cDNA for the subsequent quantitative PCR

assay. For the SARS-CoV-2 variant of concern screening in EVs, the 4Plex SC2/VOC kit (Bio-Manguinhos, Brazil) was used in a quadriplex format, employing four TaqMan probes for the simultaneous detection of the ΔH69/V70 deletion (spike), the ΔS106/G107/F108 deletion (ORF1a-nsp6), the N gene, and the internal RNase P (RP) control. Reactions were assembled according to the manufacturer's instructions and run on an ABI 7500 Fast thermocycler (Thermo Fisher Scientific), with cycle threshold (Ct) set at ≤ 40 for SARS-CoV-2 targets (WT del.69/70, WT del.nsp6, and N) and ≤ 35 for RP. Variant calls were assigned based on the pattern of deletion presence/absence: ΔH69/V70 + WT del.nsp6 for Alpha; WT del.69/70 + Δnsp6 for Beta/Gamma; WT del.69/70 + WT del.nsp6 for Delta; and ΔH69/V70 + Δnsp6 for Omicron. The interpretation of the results is shown in Table II.

*Purification of specific EVs by size exclusion chromatography (SEC)* - EV samples that tested positive for SARS-CoV-2 by RT-qPCR were pooled. Purification was initially performed by ultracentrifugation at 100,000 × g for 18 h, followed by a cross-linked agarose size exclusion chromatography with Sepharose CL-4B resin column (Merck, USA). EVs were collected through the column, separated into fractions and evaluated by chemiluminescent enzyme-linked immunosorbent assay (CL-ELISA) assay with primary antibodies marking EVs CD9 and CD81. The fractions with the best performance in detecting these markers were selected, pooled again, and subjected to a new RT-qPCR analysis, aiming at the detection of SARS-CoV-2.

*Cryo-electron microscopy (Cryo-EM)* - The size and structural integrity of each EV group from SARS-CoV-2 PCR-positive and -negative patients and healthy controls were examined by Cryo-EM on the Talos Arctica G2 Thermo Fisher, located at the National Centre for Research in Energy and Materials (CNPEM, Campinas, São Paulo, Brazil). The pooled plasma samples, represented for each EV group, were concentrated in sterile PBS and filtered after 16 h of ultracentrifugation at 100,000 × g to remove contaminants before Cryo-EM analysis. The samples containing 3 μL ($10^{10}$ to $10^{11}$ total particles) of purified EVs were applied onto the glow-discharged grids, vitrified by plunge-freezing into liquid ethane, preserving its native state, and then stored in liquid nitrogen until analysis.

TABLE I

Phenotyping of total circulating extracellular vesicles (EVs) for cellular origin (monoclonal antibodies)

| Antibodies | Fluorochromes | Cellular origin | Manufacturer |
|---|---|---|---|
| CD3 | PerCP- PE-Cy7 | T lymphocytes | BD |
| CD42a | BV421 | Platelets | BD |
| CD144 | PerCP-CY5.5 | Endothelial cells | BD |
| CD66b | FITC | Neutrophils | BD |

BD: Becton Dickinson, EUA.

TABLE II

Interpretation of 4PLEX SC2 VOC target results

| Target | Cycle threshold (Ct) values | B1.1.28, B.1.1.33, VOI Zeta (P2) and VOC** Delta | VOCs Beta and Gama (P1) | VOCs Alfa and Omicron |
|---|---|---|---|---|
| WT Del 69/70 | Ct ≤ 40.0 | D | D | ND |
| WT Del NSP6 | Ct ≤ 40.0 | D | ND | ND |
| N (Nucleocapsid) | Ct ≤ 40.0 | D | D | D |
| RP (RNAseP) | Ct ≤ 35.0 | D | D | D |

VOI: variant of interest; VOC: variant of concern; D: detected; ND: not detected.

*Statistical analysis* - The demographic and clinical data of the patients were recorded and collected on the REDCap platform. Statistical analyses of the results were performed using the GraphPad Prism 8.0 software (Graph-Pad Software, USA). Outlier exclusion from the samples was carried out using the parameters recommended by the robust regression and outlier removal (ROUT) method in the program. The normality of the variables was tested using the Shapiro-Wilk and Kolmogorov-Smirnov tests. Parametric data were analysed using the T-test and analysis of variance (ANOVA), while non-parametric data and group comparisons were analysed using the Mann-Whitney test. Statistical significance was considered at 5% ($p < 0.05$). To evaluate the performance of EVs between patients and healthy individuals, as well as for patient outcomes, we used the receiver operating characteristic (ROC) curve to define the cut-off points of the evaluated EVs. Performance variables were presented as sensitivity and specificity percentages, and the area under the curve (AUC) was used as a measure of overall accuracy.[19,20]

## RESULTS

*Demographic and clinical characteristics of the patient group* - The demographic and clinical characteristics of the 42 COVID-19 patients, presented in Table III, showed that among those comorbidities that represent a risk for worsening COVID-19, the most common were hypertension, hypothyroidism and obesity. The mean age of the patients was 50 years; the mean time of symptoms before admission was 7.6 days; and the length of hospitalisation was 13 days. All patients were admitted to the ICU with a diagnosis of COVID-19 and sepsis; sequential organ failure assessment (SOFA) scores were ≥ 2 in all patients; 85% of patients were admitted with supplemental oxygen; and 30% required vasopressor during their ICU stay. Nine patients did not survive (Table III).

*Size and concentration analysis by NTA* - NTA analysis indicated that the total EVs from 42 patients and 19 healthy donors exhibit comparable dimensions, measuring 200 nm in diameter (Fig. 1A). Conversely, particle concentrations were elevated in COVID-19 patients (1.47 x 10^10 particles/mL) compared to healthy donors (0.92 x 10^10 particles/mL) ($p < 0.05$) (Fig. 1B). No notable variations in EV concentrations were detected in relation to clinical features and outcomes among patients (Fig. 2).

*Flow cytometry* - Flow cytometry analysis of EVs from 10 healthy donors and 20 (COVID-19) patients, using the surface markers of EVs and the markers of cellular origin, show the predominance of EVs in the range of 100 nm and 240 nm, indicating a predominance of microparticles (Fig. 3).

In relation to COVID-19 patients, the markers for tetraspanins CD9 and CD81 showed mean values of 25.49% ± 5.35 and 19.91% ± 10.40, respectively, which is a significant difference (Fig. 4A). The cellular origin revealed a higher proportion of cells of lymphocytic (CD3) and platelet (CD42a) origin, followed by neutrophils (CD66b) and endothelial cells (CD144). It is worth noting that the CD42a marker showed a lower percentage value compared to CD3 due to the exclusion of outliers

from the samples (Fig. 4B). The distribution of the cellular origin of EVs did not differ between patients and healthy donors [Supplementary data (Fig. 4)].

Analyses of the cellular origin of patients EVs were conducted by comparing groups based on: gender; EV PCR-positive versus EV PCR-negative samples; duration of hospitalisation (< 7 days vs. > 7 days); mechanical ventilation; comorbidities; and differences in clinical outcomes. The percentage of tetraspanin marker (CD81) showed significant differences between patients who died and those who survived. For the clinical data regarding all other markers, no significant differences were observed [Supplementary data (Figs 5, 6, 7)].

The results of the ROC curves were evaluated using performance variables expressed as percentages (sensitivity, Se, and specificity, Sp), as well as the area under the curve (AUC), demonstrating overall accuracy.[20] ROC curves evaluating microparticles and clinical outcomes, showed an AUC of 0.929 for the number of EVs marked with the tetraspanin CD81 and clinical outcomes, death or survival (Fig. 5).

*RT-qPCR analysis* - The twenty EV samples with the highest particle concentrations by NTA were analysed by RT-qPCR using the GeneFinder kit (OSANG Healthcare, South Korea). SARS-CoV-2 RNA was detected in 14 of these samples. Some samples showed detection of all genes, others of only two genes, and others of a single gene. Considering each marker individually, the following were detected: RdRp (21%, four samples), E (26%, five samples) and N (53%, 11 samples). When the same samples were tested using the in-house AgPath one-step RT-qPCR assay (Thermo Scientific, USA), 12 yielded positive results: E (31%, five samples) and N (69%, 11 samples).

We then pooled the 14 GeneFinder-positive EV preparations, purified them by Sepharose size-exclusion chromatography, and screened the resulting fractions by ELISA for the tetraspanins CD9 and CD81. We analysed them by RT-qPCR using the GeneFinder kit (OSANG Healthcare, KOR), which showed an amplification of SARS-CoV-2 only for the N gene, with Ct ≤ 40.

Regarding variant screening with the 4Plex SC2/VOC molecular RT-qPCR kit (Bio-Manguinhos, BRA), two samples showed amplification with Ct ≤ 40 in the N

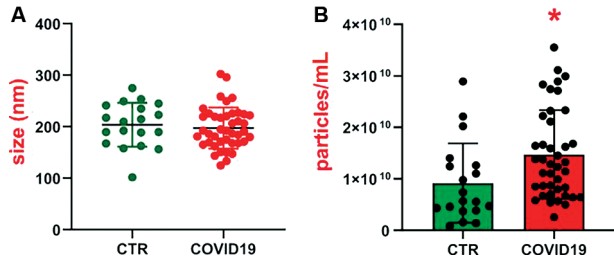

Fig. 1: extracellular vesicles (EVs) size and concentration in coronavirus disease 19 (COVID-19) patients and healthy donors, analysed by nanoparticle tracking analysis (NTA). (A) Size profile in nm of EVs. (B) Average concentrations of microparticles per particles/mL. Data analysis was performed using the non-parametric Mann-Whitney test. *p ≤ 0.05.

TABLE III

Clinical characteristics of the patient group (Coronavirus disease 19 - COVID-19)

| Characteristics | Patients (COVID-19) n = 42 | | |
|---|---|---|---|
| Age - mean (median) | 50.15 (52.5) | | |
| Male % (n) | 61.9 (26) | | |
| Days of symptoms before admission (mean) | 7.6 | | |
| Days of hospitalisation (mean) | 13.05 | | |
| Comorbidities % (n) | | | |
| Arterial hypertension | 33.3 (14) | | |
| Obesity | 23.8 (10) | | |
| Hypothyroidism | 14.28 (06) | | |
| Chronic Kidney Disease | 2.3 (01) | | |
| ICU admission data % (n) | | | |
| Supplemental oxygen | 85.7 (36) | | |
| High-flow nasal cannula | 9.5 (04) | | |
| Invasive mechanical ventilation | 4.8 (02) | | |
| SOFA score = sequential organ failure assessment % (n) | | | |
| SOFA 2 | 59.52 (25) | | |
| SOFA 3 | 14.28 (06) | | |
| SOFA 4 | 16.66 (07) | | |
| SOFA 5, 6 and 8 | 9.52 (04) | | |
| Laboratory parameters on admission expressed | Mean ± SD (n) | Median (Q1-Q3) | Reference values (M) Male, (F) Female |
| Haemoglobin (g/dL) | 13.79 ± 1.77 (42) | 13.8 (12.38-15.05) | (M)13.5-17.5 (F)12.0-16.0 |
| Leukocytes (×10³/mm³) | 10.05 ± 5.07 (42) | 9.67 (6.55-12.42) | 4.00-11.00 |
| Neutrophils (×10³/mm³) | 7.754 ± 4.00 (42) | 7.24 (4.94-9.56) | 1.50-8.00 |
| Lymphocytes (×10³/mm³) | 1.20 ± 0.76 (42) | 1.12 (0.73-1.53) | 1.00-4.00 |
| Platelets (×10³/mm³) | 223.40 ± 79.54 (42) | 207.5 (166.8-262.8) | 150-450 |
| Creatinine (mg/dL) | 1.15 ± 0.80 (42) | 0.91 (0.80-1.13) | (M) 0.7-1.3 (F) 0.6-1.1 |
| Glucose (mg/dL) | 162.53 ± 64.37 (40) | 147.5 (122.5-186.0) | 70-99 |
| C-Reactive Protein (mg/L) | 14.92 ± 8.10 (41) | 14.19 (8.01-20.21) | < 3.0 |
| Lactate (mg/dL) | 13.27 ± 5.03 (30) | 13.0 (9.75-16.0) | 4.5-19.8 |
| ALT (TGP) (U/L) | 69.46 ± 59.32 (35) | 47.0 (36.0-78.0) | (M) up to 41 (F) up to 33 |
| AST (TGO) (U/L) | 62.60 ± 56.06 (35) | 46.0 (36.0-74.0) | up to 40 |
| Clinical support during ICU stay % (n) | | | |
| Invasive mechanical ventilation | 35.71 (15) | | |
| Use of vasopressors | 30.9 (13) | | |
| Outcome % (n) | | | |
| Hospital discharge | 78.57 (33) | | |
| Death | 21.42 (09) | | |

ALT (TGP): alanine aminotransferase (transaminase glutamate - pyruvate); AST (TGO): aspartate aminotransferase (transaminase glutamate - oxaloacetate); SD: standard deviation.

genes, the RP target Ct ≤ 35 in the WT del69/70 deletion, and no detection of NSP6 in WT deletion, suggesting the Gamma variant (Fig. 6A). In the analysis of the average concentrations of microparticles (particles/mL) by NTA of PCR-positive EVs versus PCR-negative EVs, there was no significant difference between the groups (Fig. 6B).

In the flow cytometry analysis, there were no significant differences in the concentrations (EVs/mm³) of:

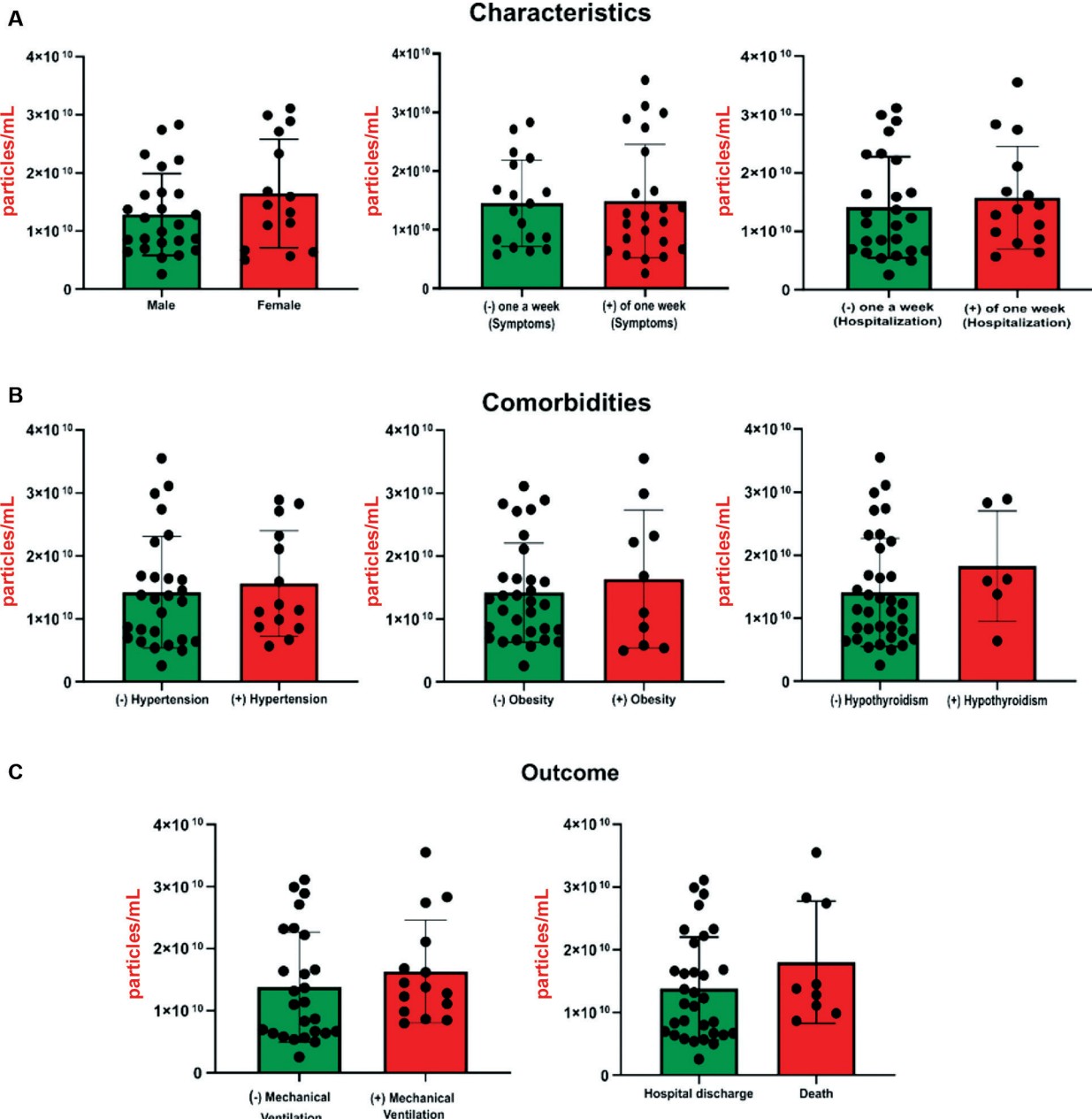

Fig. 2: extracellular vesicles (EVs) concentrations according to coronavirus disease 19 (COVID-19) patients characteristics and clinical course. EVs were measured by nanoparticle tracking analysis (NTA) and expressed as particles/mL. Data analysis was performed using the non-parametric Mann-Whitney test. Significant differences were considered with a p-value < 0.05 and highlighted with (*).

cellular expression with the phosphatidylserine marker (Annexin V); the percentages of cellular expression with tetraspanin markers (CD9 and CD81); and cellular origin markers (CD42a, CD66b, CD3 and CD144) in PCR-positive EVs versus PCR-negative EVs (Fig. 6C).

*Cryo-electron microscopy* - We used Cryo-EM, which allows for the visualisation of EVs while preserving their membranes in a near-native state, to reveal the morphology and size of the EVs. We obtained images of EVs from samples that tested positive and negative in RT-qPCR for SARS-CoV-2 genes, as well as EVs from healthy donors, showing morphological similarities within each group.

Most EVs had a rounded shape (Fig. 7A, B, C). The complete images (Fig. 7A1, B2, C2), approximately 200 nm in size, show a distinct outer layer, the lipid bilayer characteristic of EVs.[21,22] Inside, the internal phase of the vesicles is evident, which potentially includes intracellular materials such as proteins, RNA or other biomolecules. Unfortunately, we were not able to determine or visualise the presence of SARS-CoV-2 in the images.

## DISCUSSION

Previous studies have revealed that EVs released by virus-infected cells carry viral RNA, as in the case of HIV, HCV, influenza and others.[23] Indeed, the interac-

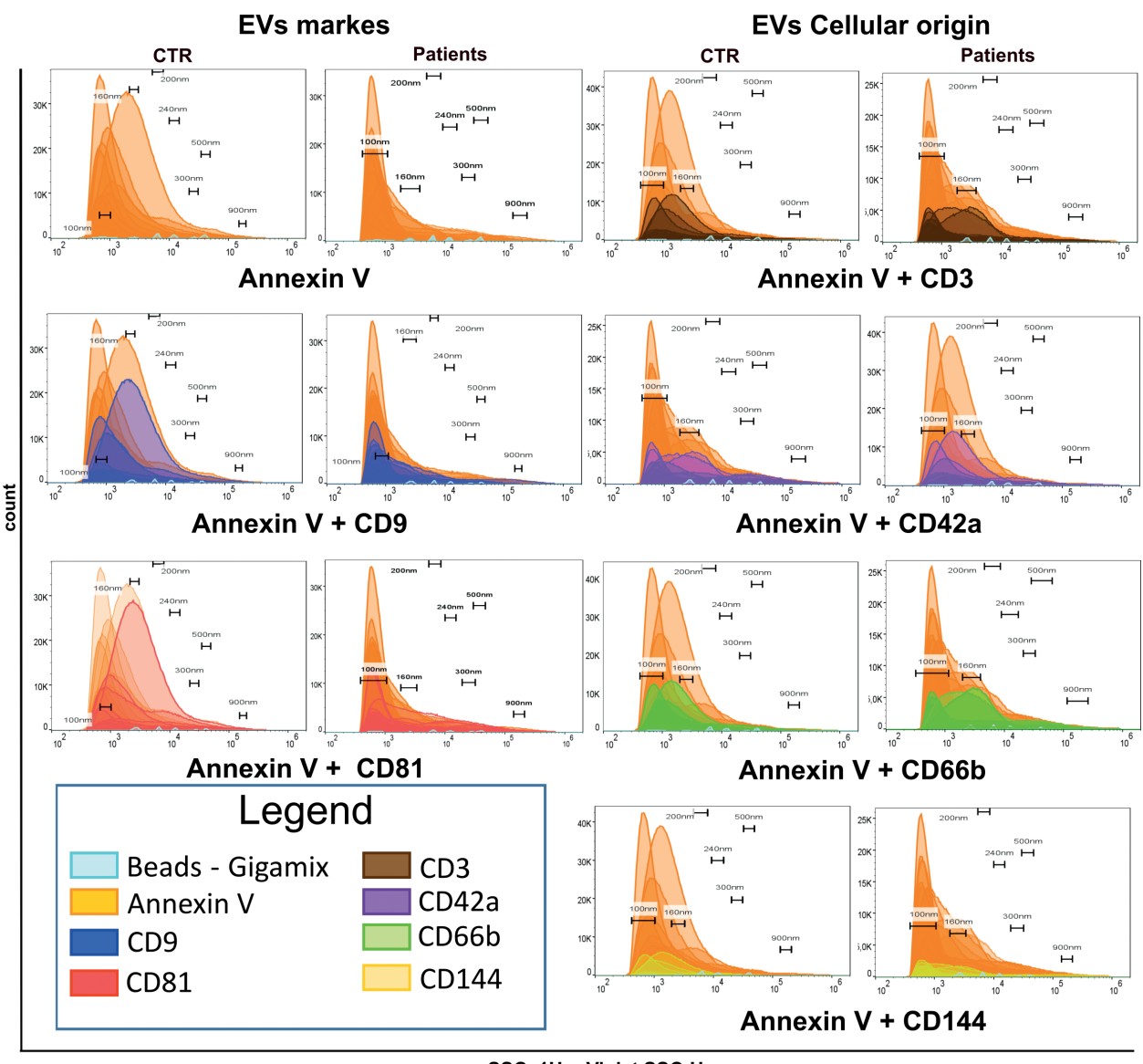

Fig. 3: overlays of one-dimensional histograms of gigamix and extracellular vesicles (EVs) ranging from 100 nm to 900 nm. Comparison between healthy volunteers (CTR) and patients for each marker.

tions between EVs and viruses are highly interconnected and involve viral propagation and immune regulation, and there are hypotheses regarding their potential for use as biomarkers for infection diagnosis.[24]

The interaction between SARS-CoV-2 and EVs has generated increasing interest, although it has not been fully understood yet. Using RT-qPCR, we detected viral components in EV samples from 14 patients using the GeneFinder kit, and in 12 patients using the in-house AgPath One-Step kit. In our results, both kits showed a predominance of N gene detection in EVs, with 53% for the GeneFinder kit and 69% for the in-house AgPath One-Step kit.

Using RT-qPCR with the GeneFinder kit, we analysed a "pool" of the 14 EVs that showed amplification of SARS-CoV-2 genes. These EVs were purified using a Sepharose column and the best fractions expressing the CD9 and CD81 markers were selected. This confirmed SARS-CoV-2 amplification, but only for the N gene, with a Ct ≤ 40. In an article published a few months after the start of the pandemic in 2020, Chu DK et al.[25] evaluated the sensitivity of RT-PCR detection kits and indicated that the N gene is the most sensitive for detecting SARS-CoV-2 in clinical samples. Furthermore, in a study published in 2022, despite the emergence of new variants, SARS-CoV-2 was detected in swab samples from the nasopharynx and pharynx.[25] This study demonstrated that the ORF1ab and N genes had the best sensitivity when compared to samples confirmed as positive, especially in samples with low viral titers.[26]

The Alpha, Gamma and Omicron variants exhibit the R203K/G204R mutation in the SARS-CoV-2 N protein, contributing to increased viral replication.[27,28] Another important factor is that the N protein, with its highly con-

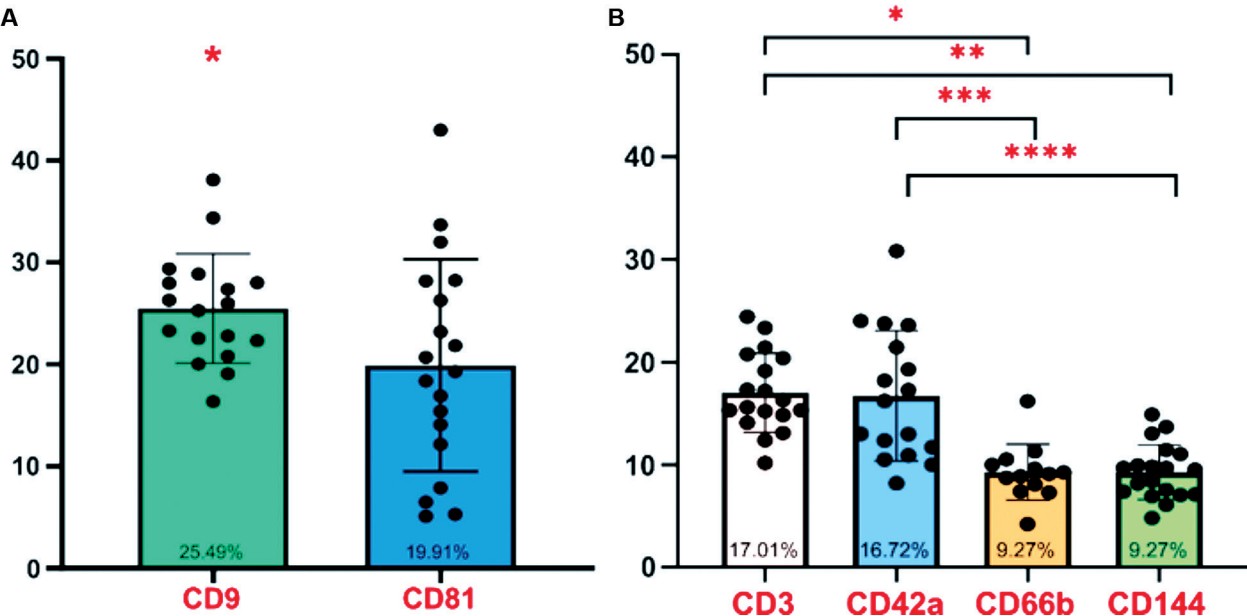

Fig.4: percentages of extracellular vesicles (EVs) markers and cellular origin in patients with coronavirus disease 19 (COVID-19). (A) Mean percentage expression of vesicular markers, tetraspanins, within the gate marked with Annexin V (Phosphatidylserine). The data were analysed using the parametric T-test. (B) Mean percentage expression of cellular origin markers within the gate marked with Annexin V (Phosphatidylserine). The data was analysed using the parametric analysis of variance (ANOVA) test. Significant differences ($p < 0.05$) are indicated with (*).

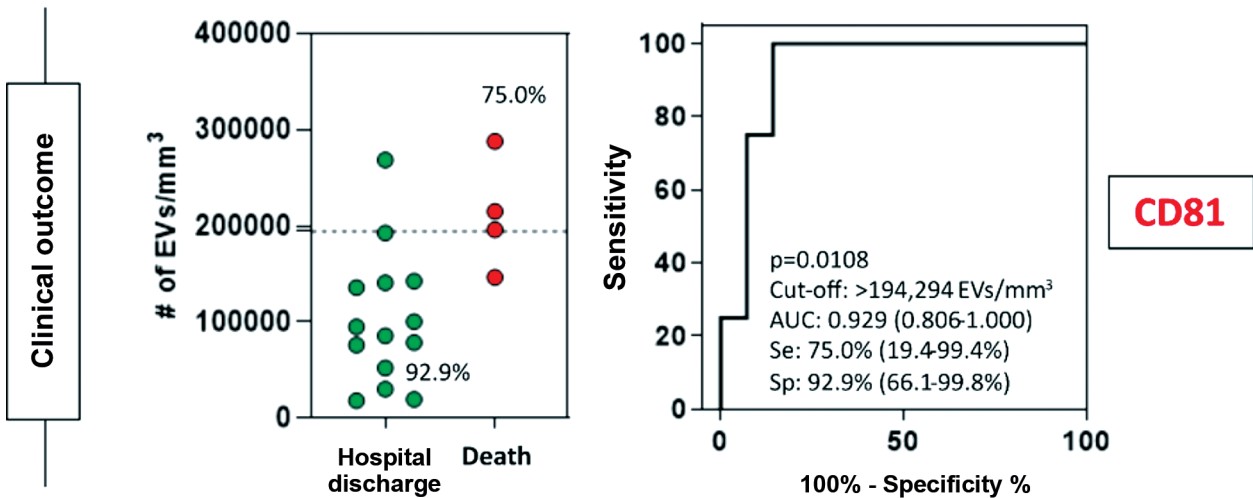

Fig. 5: receiver operating characteristic (ROC) curve evaluating the performance of extracellular vesicles (EVs) with the tetraspanin marker (CD81) in differentiating clinical outcomes between hospital discharge and death. The curves display cut-off limits, areas under the curve (AUC), sensitivity (Se), and specificity (Sp) values. In the scatter plots, the dashed lines represent the cut-off limits and the inserted proportions indicate the Se and Sp values. Significant differences were identified with a p-value < 0.05.

served N-terminal (NTD) and C-terminal (CTD) domain structures, provides stability in its three-dimensional conformation, making it a stable protein that facilitates specific binding to viral RNA and interaction with other viral proteins.[28] We can hypothesise that a higher prevalence of viral components, particularly the N, in EVs may indicate a role in promoting viral dissemination to other cells. Using a previously described PCR variant screen-

ing strategy we were able to characterise two EV samples as belonging to the prevalent Gamma variant.

The presence of SARS-CoV-2 particles in EV samples aligns with previous reports. Sun et al.[29] showed the presence of viral particles near EVs isolated from sputum supernatants of COVID-19 patients via electron microscopy. They demonstrated that exosomes extracted from SARS-CoV-2-infected VeroE6 cells expressed

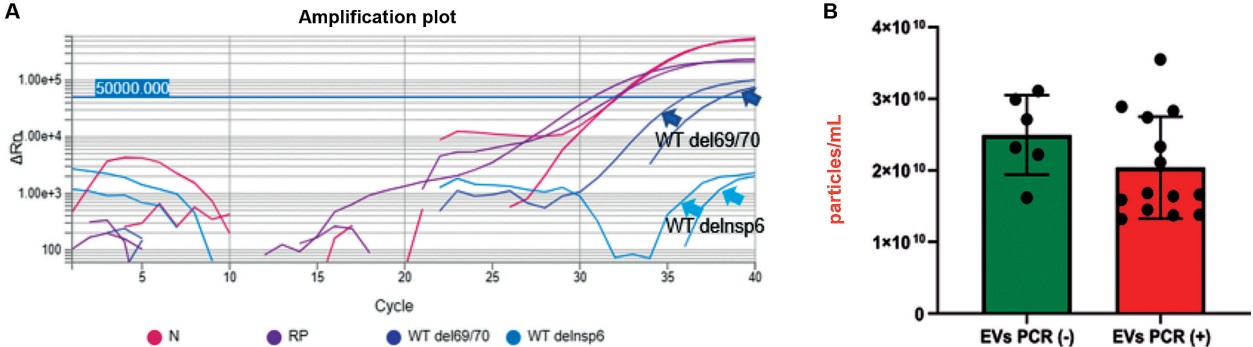

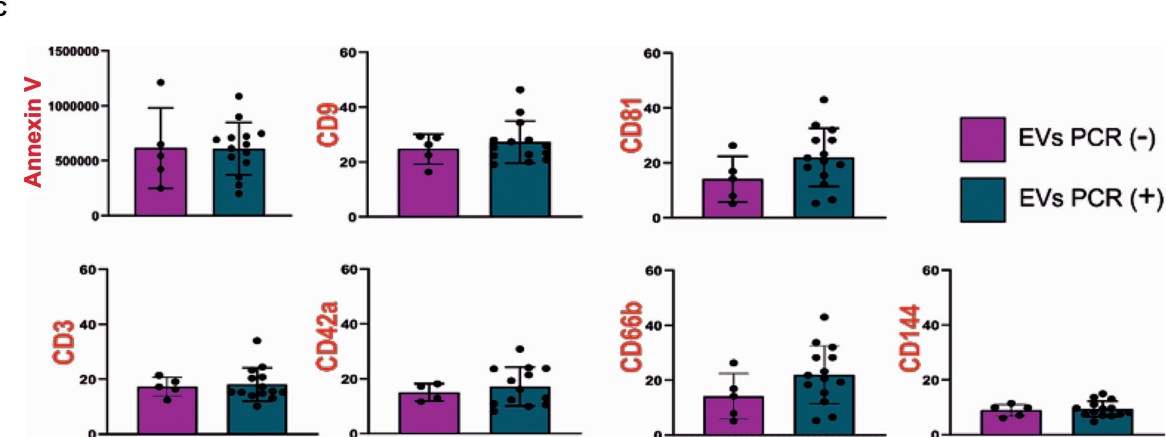

Fig. 6: polymerase chain reaction (RT-qPCR) analysis of isolated extracellular vesicles (EVs). (A) RT-qPCR amplification graph using the 4PLEX SC2 VOC kit for variant screening of two samples showed amplification with cycle threshold (Ct) ≤ 40.0 in the Nucleocapsid (N) and RNAseP (RP) genes, highlighting the WT del69/70 deletion and the non-detection of the WT del Nsp6 deletion gene, suggesting the Gamma variant. (B) Comparative graph in the nanoparticle tracking analysis (NTA) showing the average concentrations of microparticles (particles/mL) in PCR-positive EVs versus PCR-negative EVs. Data analysis was performed using the non-parametric Mann-Whitney test. (C) Comparative graph in the flow cytometry analysis of PCR-positive EVs versus PCR-negative EVs. Data analysis was performed using the parametric T-test. Significant differences ($p < 0.05$) were indicated with (*).

the N and spike (S) proteins of SARS-CoV-2, suggesting that exosomes are involved in viral transmission. Another *in vitro* study isolated SARS-CoV-2-infected VeroE6 cells, revealing the presence of SARS-CoV-2 N and S proteins in EVs.[29,30]

Our EV analyses via Cryo-EM revealed similar results, showing a predominance of EVs with well-defined lipid bilayers in the membranes. Most EVs had a rounded shape and an average size of 200 nm. This finding aligns with previous reports. Emelyanov et al.,[31] evaluating EV morphology in cerebrospinal fluid samples via Cryo-EM, found that more than 80% of EVs had a lipid bilayer/ membrane and a rounded shape. The remaining EVs had an elongated shape.[31] When analysing plasma samples from healthy donors with isolated EVs in 2014, Arraud et al.[21] obtained Cryo-EM images showing rounded EVs with a well-defined outer layer in two lines.[21]

According to Zhu et al.,[32] the SARS-CoV-2 virus has an average size ranging from 60 to 140 nm. We suggest that the EVs we isolated are potential carriers of viral components or even the entire virus. Additional imaging analyses with viral markers are necessary to confirm the presence of viral components.[32] NTA can track individual particles and estimate their size, concentration and behaviour in real-time. However, it must be interpreted cautiously due to the presence of co-isolated particles, such as lipoproteins, protein aggregates and EVs.[33,34] To encompass co-isolates, we use the term "microparticles." Regarding the comparative concentration between healthy individuals and patients with COVID-19 and sepsis, a significantly higher concentration of microparticles was observed in patients based on NTA analyses. In terms of clinical outcomes among patients, no significant differences were identified between groups.

For EV characterisation via flow cytometry, we selected cellular origin markers and Annexin V for the detection of phosphatidylserine, along with tetraspanin markers CD9 and CD81, to confirm the presence of isolated EVs. Annexin V is a protein that binds to phosphatidylserine and is found on the inner face of the plasma membrane in cells. When a cell undergoes apoptosis or secretes EVs, phosphatidylserine may externalise to the outer face of the membrane.[22,35] Tetraspanins, which are involved in EV biogenesis, are a family of transmembrane proteins that organise cellular membrane microdomains.[36,37] Flow cytometry revealed a predominance of

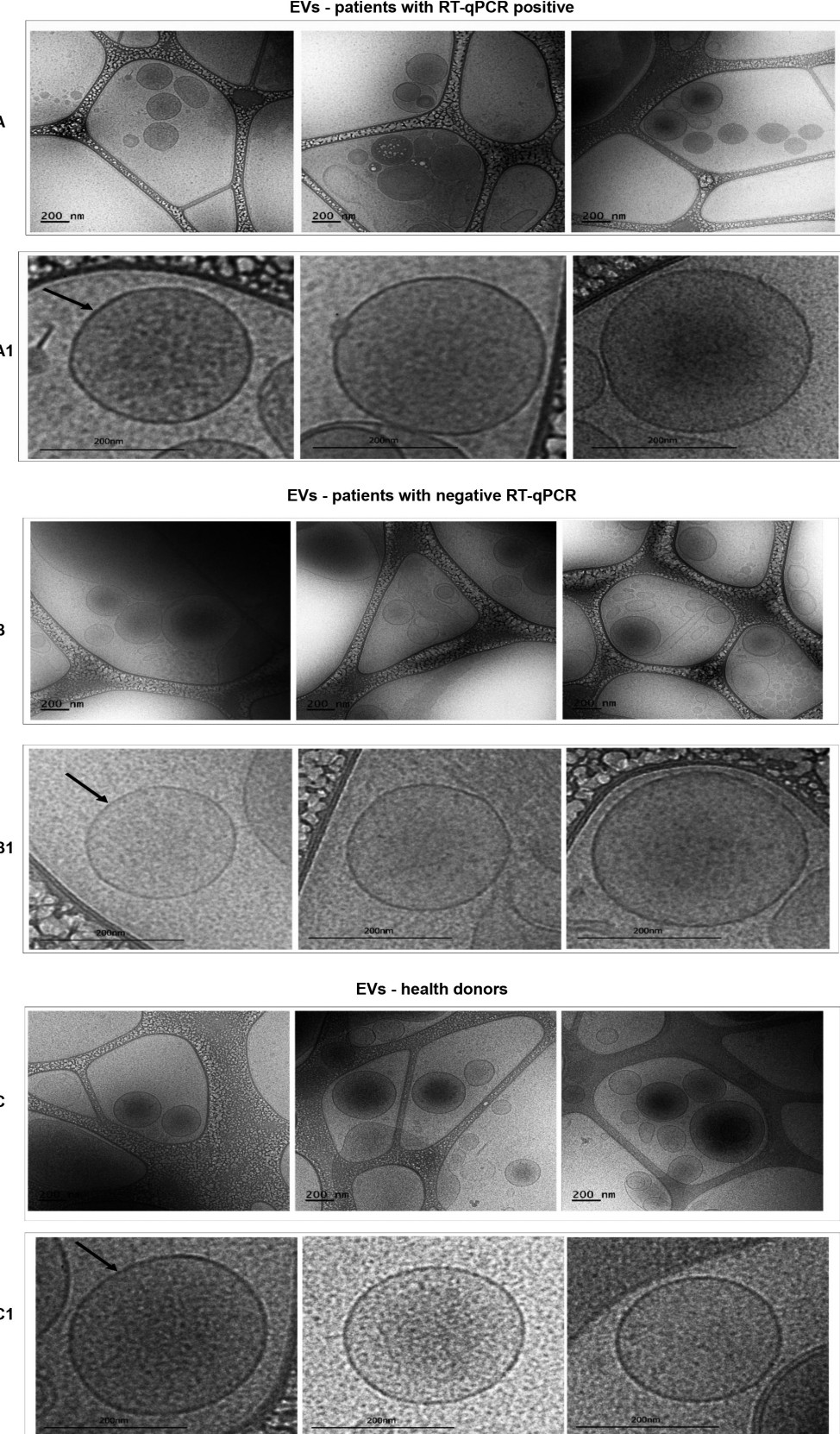

Fig. 7: images of total extracellular vesicles (EVs) obtained using cryo-electron microscopy (Cryo-EM). (A) EVs isolated from severe acute respiratory syndrome Coronavirus 2 (SARS-CoV-2) positive samples. (B) EVs isolated from polymerase chain reaction (RT-qPCR) negative patient samples. (C) EVs isolated from healthy donor samples. (A1, B1 and C1) Zoomed-in views of the images for each group. Arrows indicate the lipid bilayer of the EVs.

EVs in the 100-240 nm range. This size range is consistent with NTA data and Cryo-EM visualisation. Based on the EV size classification published by the ISEV[22] we suggest the presence of exosomes, but with a predominance of microvesicles. Poveda E et al.[38] evaluated microvesicles that were isolated from plasma samples of HIV-infected patients and uninfected controls. The microvesicles ranged from 300-1000 nm and expressed the tetraspanin CD9 via flow cytometry.[38] Our results show the expression of tetraspanins CD9 and CD81, with a higher expression of CD9. The markers CD9 and CD81 are excellent choices for the identification of exosomes and microvesicles.

We were interested in evaluating the cellular origin of EVs that were present in plasma samples from our patients. Peripheral blood leukocytes were altered in COVID-19 patients, and neutrophilia and lymphopenia are predictive of disease severity and can identify patients at risk of severe complications.[39,40] Blood test data upon hospital admission showed normal lymphocyte and platelet levels and elevated neutrophil levels in our cohort. The cellular origin of EVs isolated from plasma samples revealed a higher proportion of lymphocyte- and platelet-derived EVs, both significantly higher compared to neutrophil- and endothelial cell-derived EVs. Nevertheless, we conclude that the EVs were heterogeneous.

EV concentration by flow cytometry, cellular origin, and characterisation showed no significant differences between the patient and healthy control groups. Our data support the work of Li P et al.,[41] who found no significant differences in the levels of circulating EVs of platelet, endothelial cell or leukocyte origin between septic patients and healthy individuals.[41]

Overall, no significant differences were observed for each cellular marker regarding epidemiological and clinical outcomes among patients, except for the increased expression of the endothelial cell-origin marker (CD144) in patients not subjected to mechanical ventilation. In addition, it is important to note that the CD81 tetraspanin showed increased expression in deceased patients, with a ROC curve showing 75% sensitivity and 93% specificity for outcomes. This study indicates that the CD81 tetraspanin marker may differentiate between discharged and deceased patients.

EVs are biological structures that carry a wide range of molecules, including mRNAs, microRNAs and other non-coding RNAs, which play crucial roles in the progression and modulation of various pathological processes. Recent evidence indicates that the molecular composition of EVs can vary according to the pathophysiological context, thereby reflecting the state of the parental cell and influencing disease evolution.[42,43,44,45] Particular attention has been devoted to EVs transporting viral RNA, as these vesicles can activate the immune system, modulate gene expression, interfere with cellular functions and facilitate infectious processes, thereby promoting intercellular communication across tissues and distances.[46,47] Such mechanisms of EV-mediated RNA transfer are being investigated extensively not only to elucidate the potential of vesicle-associated viral RNA to initiate new replication cycles,[48] but also to get a better understanding of their involvement in the clinical course of viral infections. Moreover, the identification of specific EV-associated biomarkers has emerged as a promising strategy for the development of innovative diagnostic approaches. [49,50] Regarding microRNAs, EVs containing specific miRNA cargo also rely on these communication mechanisms to exert relevant regulatory functions. Notably, these vesicles have been reported to reduce mortality in certain pathological contexts[51] and can confer cardioprotective effects in animal models of sepsis,[52] highlighting their therapeutic potential.

Our study has some limitations, such as the small number of patients (42, all from a single medical institution) and the difficulty in obtaining sufficient volumes of EVs. In addition, flow cytometry, RT-qPCR and electron microscopy analyses were performed on a subgroup of 20 patients. On the other hand, our study has strengths, such as the prospective cohort and the broad evaluation of EVs by NTA, flow cytometry and Cryo-EM.

In summary, our findings indicate that patients in an ICU with COVID-19 and sepsis exhibit a higher concentration of EVs compared to healthy controls, characterised by a size of 200 nm and a well-defined lipid bilayer in their membranes. An RT-qPCR analysis of EVs demonstrated positive detection of viral components, indicating their potential role as carriers of these components, which may facilitate viral propagation and enhance the susceptibility of healthy cells to infection. CD81 tetraspanin has the potential to distinguish between surviving and non-surviving patients, but this finding requires validation in larger cohorts.

## ACKNOWLEDGEMENTS

To the Brazilian Nanotechnology National Laboratory (LNNano) — part of the Brazilian Centre for Research in Energy and Materials (CNPEM) —, a private non-profit organisation under the supervision of the Brazilian Ministry for Science, Technology and Innovation (MCTI). The CRYO-EM staff is acknowledged for their assistance during the experiments (20230431).

## AUTHORS' CONTRIBUTION

JFNC - sample selection, EV isolation, flow cytometry analysis, RT-qPCR analysis, sample separation for Cryo-EM analysis, statistical analysis, writing and revision of the manuscript; PM - EV charcterisation by NTA, EV isolation, purification by SEC, CL-ELISA and sample separation for Cryo-EM analysis; GRB - SARS-CoV2 RT-qPCR analysis; MMAS - flow cytometry analysis; SSS - flow cytometry analysis; FF - clinical investigation and cohort coordinator; DB - patient enrolment and sampling; NCJB - SARS-CoV2 RT-qPCR analysis; ATC - flow cytometry analysis and statistical analysis; ACT - Project co-supervisor, EV isolation, analysis of EVs concentration and size, sample separation for Cryo-EM analysis, conceptualisation, writing and revision of the manuscript; funding acquisition; RS - conceptualisation, project supervisor (clinical and laboratory steps), funding acquisition, writing and revision of the article. All authors have revised the final version of the manuscript.

## DATA AVAILABILITY

The contents underlying the research text are included in the manuscript.

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

# OPEN PEER REVIEW

Memórias do IOC thanks the anonymous reviewers for their contribution to the peer review of this work.

## FIRST REVIEW ROUND

REVIEWERS' COMMENTS

**REVIEWER #1**

Reviewer comments: The manuscript presents a well-designed study investigating the role of extracellular vesicles (EVs) in COVID-19 and sepsis, with a focus on their characterization, viral cargo, and clinical correlations. The study is timely and addresses an important gap in understanding the pathophysiology of severe COVID-19. The methodologies are robust, combining nanoparticle tracking analysis (NTA), flow cytometry, RT-qPCR, and cryo-EM. However, some aspects require clarification, and additional references would strengthen the discussion. Below are specific comments organized by section.

a) Abstract Adequacy

The abstract provides a concise overview of the study's objectives and key findings. It clearly states the investigation of EV characteristics in COVID-19 and sepsis patients, including their size, concentration, and viral cargo. However, it could be strengthened by briefly mentioning the clinical implications of the findings, particularly the association between CD81+ EVs and patient outcomes. Adding a sentence about the potential diagnostic or prognostic value of these EVs would enhance its impact.

b) Originality and Importance

This study makes several significant contributions to the field. First, it provides novel evidence that EVs from COVID-19 patients can carry SARS-CoV-2 RNA, including the Gamma variant, which has not been extensively reported. Second, the identification of CD81+ EVs as a potential biomarker for mortality is clinically relevant and could inform future prognostic tools. The study also advances technical methodologies by combining NTA, flow cytometry, and cryo-EM for comprehensive EV characterization. These findings are particularly important given the ongoing need to understand the pathophysiology of severe COVID-19 and sepsis, as well as the role of EVs in these conditions.

c) Methodology, Results, and Discussion

The methodology is robust, employing multiple techniques to characterize EVs, including NTA for size and concentration, flow cytometry for cellular origin, and RT-qPCR for viral RNA detection. The use of cryo-EM to visualize EVs adds valuable morphological data. However, the study could benefit from additional details on EV isolation protocols, such as steps taken to minimize lipoprotein contamination, and clarification on whether isotype controls were used in flow cytometry experiments to ensure antibody specificity.

The results are well-presented and demonstrate clear differences in EV profiles between patients and controls. The detection of viral RNA in EVs, particularly the nucleocapsid gene, is a notable finding. However, the discussion could be expanded to explore potential mechanisms by which EV-associated viral RNA might contribute to disease progression, such as through immune activation or endothelial dysfunction. Comparing these findings to existing literature on EVs in bacterial sepsis would also provide broader context.

d) References

The reference list is generally appropriate but could be strengthened by including additional key studies. For example, recent work on EV-mediated endothelial injury in COVID-19 (DOI: 10.1002/jev2.12117) and the role of EVs in sepsis-associated ARDS (DOI: 10.3389/fimmu.2023.1150564) would provide valuable context. Additionally, citing the MISEV guidelines (DOI: 10.1002/jev2.12404) would underscore the study's adherence to current EV research standards.

e) Figures and Tables

The figures and tables are clear and effectively support the study's findings. Figure 1 provides a good visualization of EV size and concentration differences between groups, while Figure 5's ROC curve effectively highlights the prognostic value of CD81+ EVs. However, annotating the cryo-EM images to highlight specific features, such as double-membrane structures or putative viral particles, would enhance their interpretability. Table III is comprehensive but could include interquartile ranges for laboratory values to better represent data variability.

Overall Strengths
- Comprehensive multi-method approach to EV characterization.
- Novel findings on EV-associated viral RNA and its clinical implications.
- Strong statistical analysis, including ROC curves for prognostic evaluation.

Areas for Improvement
- Provide more details on EV isolation and purity controls.
- Expand the discussion to include mechanistic hypotheses and comparisons with bacterial sepsis.
- Update references to include recent studies on EVs in COVID-19 and sepsis.

Recommendation
This manuscript presents valuable findings that advance our understanding of EVs in COVID-19 and sepsis. With minor revisions to address the points above, it would be suitable for publication.

## AUTHORS' RESPONSE TO THE REVIEWERS

Reviewer 1:
Reviewer's comments: The manuscript presents a well-designed study investigating the role of extracellular vesicles (EVs) in COVID-19 and sepsis, with a focus on their characterization, viral cargo, and clinical correlations. The study is timely and addresses an important gap in understanding the pathophysiology of severe COVID-19. The methodologies are robust, combining nanoparticle tracking analysis (NTA), flow cytometry, RT-qPCR, and cryo-EM. However, some aspects require clarification, and additional references would strengthen the discussion. Below are specific comments organized by section.

ANSWER: We would like to thank you for the critical analysis of our manuscript. We added all suggestions to improve the manuscript for publication. The corrections on the manuscript were in highlight in yellow.

Reviewer's comments:

a. Abstract Adequacy
The abstract provides a concise overview of the study's objectives and key findings. It clearly states the investigation of EV characteristics in COVID-19 and sepsis patients, including their size, concentration, and viral cargo. However, it could be strengthened by briefly mentioning the clinical implications of the findings, particularly the association between CD81+ EVs and patient outcomes. Adding a sentence about the potential diagnostic or prognostic value of these EVs would enhance its impact.

ANSWER: We do agree with the referee. We aimed to include this information, while keeping the abstract restrict to 200 words, as requested by the MIOC.

Lines 42 to 51- "RESULTS: NTA showed an increased concentration of microparticles in patients. RT-qPCR analysis of EVs detected the virus in 14 samples, two of which were consistent with the Gamma variant. Extracellular vesicles (EVs) predominantly derived from T cells and platelets, with increased CD81 expression in individuals who died. Cryo-EM revealed extracellular vesicles with an average size of 200 nm. CONCLUSIONS: Our findings suggest that patients' extracellular vesicles likely harbour viral components, suggesting their potential role as carriers of SARS-CoV-2, and that extracellular vesicles from deceased patients demonstrate elevated CD81 expression."

Originality and Importance.
This study makes several significant contributions to the field. First, it provides novel evidence that EVs from COVID-19 patients can carry SARS-CoV-2 RNA, including the Gamma variant, which has not been extensively reported. Second, the identification of CD81+ EVs as a potential biomarker for mortality is clinically relevant and could inform future prognostic tools. The study also advances technical methodologies by combining NTA, flow cytometry, and cryo-EM for comprehensive EV characterization. These findings are particularly important given the ongoing need to understand the pathophysiology of severe COVID-19 and sepsis, as well as the role of EVs in these conditions.

ANSWER: We thank the referee for the comment.

Methodology, Results, and Discussion
The methodology is robust, employing multiple techniques to characterize EVs, including NTA for size and concentration, flow cytometry for cellular origin, and RT-qPCR for viral RNA detection. The use of cryo-EM to visualize EVs adds valuable morphological data. However, the study could benefit from additional details on EV isolation protocols, such as steps taken to minimize lipoprotein contamination, and clarification on whether isotype controls were used in flow cytometry experiments to ensure antibody specificity.

ANSWER. We described the protocol "*Purification of specific EVs by size exclusion chromatography (SEC)*" in the Lines 222 to 230. The Sepharose CL-4B column increased the size exclusion separation of extracellular vesicles (EVs), efficiently removing soluble proteins and smaller lipoproteins, including LDL (20–30 nm), VLDL (30–80 nm), and HDL (7–12 nm). Consequently, samples enriched in extracellular vesicles were acquired, exhibiting most particles in size ranges above those of these lipoproteins. Subsequent analysis via NTA and cryo-electron microscopy verified that nearly 95% of the isolated particles were above the diameters associated with these contaminants.

Results section
*We add Cryo-Electron Microscopy (Cryo-EM).* We used cryo-electron microscopy (Cryo-EM), which allows

for the visualization of EVs while preserving their membranes in a near-native state, revealing the morphology and size of the EVs. We obtained images of EVs from samples that tested positive and negative in RT-qPCR for SARS-CoV-2 genes, as well as EVs from healthy donors, showing morphological similarities within each group. Most EVs had a rounded shape (Fig. 7 A, B, and C). The complete images (Fig. 7 A1, B2, and C2), approximately 200 nm in size, show a distinct outer layer, the lipid bilayer characteristic of extracellular vesicles (29,33); inside, the internal phase of the vesicles is evident, potentially which includes intracellular materials such as proteins, RNA, or other biomolecules. Unfortunately, we are not to determine or visualize the presence of SARS-CoV-2 in the images

Discussion

Our EV analyses via Cryo-EM revealed similar results, showing a predominance of EVs with well-defined lipid bilayers in the membranes. Most EVs had a rounded shape and an average size of 200 nm. This finding aligns with previous reports. Emelyanov et al. (2020), evaluating EV morphology in cerebrospinal fluid samples via cryo-electron microscopy (Cryo-EM), found that more than 80% of EVs had a lipid bilayer/membrane and a rounded shape. The remaining EVs had an elongated shape (28). Previously, in 2014, Arraud et al., when analyzing plasma samples from healthy donors with isolated EVs, obtained Cryo-EM images showing EVs with a well-defined outer layer in two lines, round-shaped EVs (29).

(comments): The results are well-presented and demonstrate clear differences in EV profiles between patients and controls. The detection of viral RNA in EVs, particularly the nucleocapsid gene, is a notable finding. However, the discussion could be expanded to explore potential mechanisms by which EV-associated viral RNA might contribute to disease progression, such as through immune activation or endothelial dysfunction. Comparing these findings to existing literature on EVs in bacterial sepsis would also provide broader context.

ANSWER. We added in the discussion. "Extracellular vesicles (EVs) are biological structures that carry a wide range of molecules, including mRNAs, microRNAs, and other non-coding RNAs, which play crucial roles in the progression and modulation of various pathological processes. Recent evidence indicates that the molecular composition of EVs can vary according to the pathophysiological context, thereby reflecting the state of the parental cell and influencing disease evolution (41; 42; 43; 44). Particular attention has been devoted to EVs transporting viral RNA, as these vesicles can activate the immune system, modulating gene expression, interfering with cellular functions, and facilitating infectious processes, thereby promoting intercellular communication across tissues and distances (45; 46). Such mechanisms of EV-mediated RNA transfer are being extensively investigated not only to elucidate the potential of vesicle-associated viral RNA to initiate new replication cycles (47), but also to better understand their involvement in the clinical course of viral infections. Moreover, the identification of specific EV-associated biomarkers emerges as a promising strategy for the development of innovative diagnostic approaches (48; 49). Regarding microRNAs, EVs containing specific miRNA cargo also rely on these communication mechanisms to exert relevant regulatory functions. Notably, these vesicles have been reported to reduce mortality in certain pathological contexts (50) and to confer cardioprotective effects in animal models of sepsis (51), highlighting their therapeutic potential."

d) References

The reference list is generally appropriate but could be strengthened by including additional key studies. For example, recent work on EV-mediated endothelial injury in COVID-19 (DOI: 10.1002/jev2.12117) and the role of EVs in sepsis-associated ARDS (DOI: 10.3389/fimmu.2023.1150564) would provide valuable context. Additionally, citing the MISEV guidelines (DOI: 10.1002/jev2.12404) would underscore the study's adherence to current EV research standards.

ANSWER In this study, we employed standardized methodologies for extracellular vesicle (EV) isolation and characterization. Specifically, ultracentrifugation and column-based particle separation were performed, following the guidelines established by the Minimal Information for Studies of Extracellular Vesicles (MISEV 2023, reference number 32. Welsh J, Goberdhan DC, O'Driscoll L, Buzas EI, Blenkiron C, Bussolati B, et al. Minimal information for studies of extracellular vesicles (MISEV2023): from basic to advanced approaches. J Extracell Vesicles. 2024;13(2):e12404. doi:10.1002/jev2.12404 in the Manuscript's references). These approaches were selected to ensure methodological rigor, reproducibility, and compliance with current international recommendations. Furthermore, the inclusion of these procedures and their detailed citation were carried out in accordance with the reviewer's suggestions, thereby strengthening the methodological robustness and alignment with the most up-to-date consensus in the field.

e) Figures and Tables

The figures and tables are clear and effectively support the study's findings. Figure 1 provides a good visualization of EV size and concentration differences between groups, while Figure 5's ROC curve effectively highlights the prognostic value of CD81+ EVs. However, annotating the cryo-EM images to highlight specific features, such as double-membrane structures or putative viral particles, would enhance their interpretability. Table III is comprehensive but could include interquartile ranges for laboratory values to better represent data variability.

ANSWER. We added arrows to the Cryo-EM images to indicate the lipid bilayer of the EVs in the samples from patients with positive and negative RTq PCR and healthy donors.

*Protocol -Cryo-Electron Microscopy (Cryo-EM).* We used cryo-electron microscopy (Cryo-EM), which allows for the visualization of EVs while preserving their membranes in a near-native state, revealing the morphology and size of the EVs. We obtained images of EVs from samples that tested positive and negative in RT-qPCR for SARS-CoV-2 genes, as well as EVs from healthy donors, showing morphological similarities within each group. Most EVs had a rounded shape (Fig. 7 A, B, and C). The complete images (Fig. 7 A1, B2, and C2), approximately 200 nm in size, show a distinct outer layer, the lipid bilayer characteristic of extracellular vesicles (29,33); inside, the internal phase of the vesicles is evident, potentially which includes intracellular materials such as proteins, RNA, or other biomolecules. Unfortunately, we are not to determine or visualize the presence of SARS-CoV-2 in the images

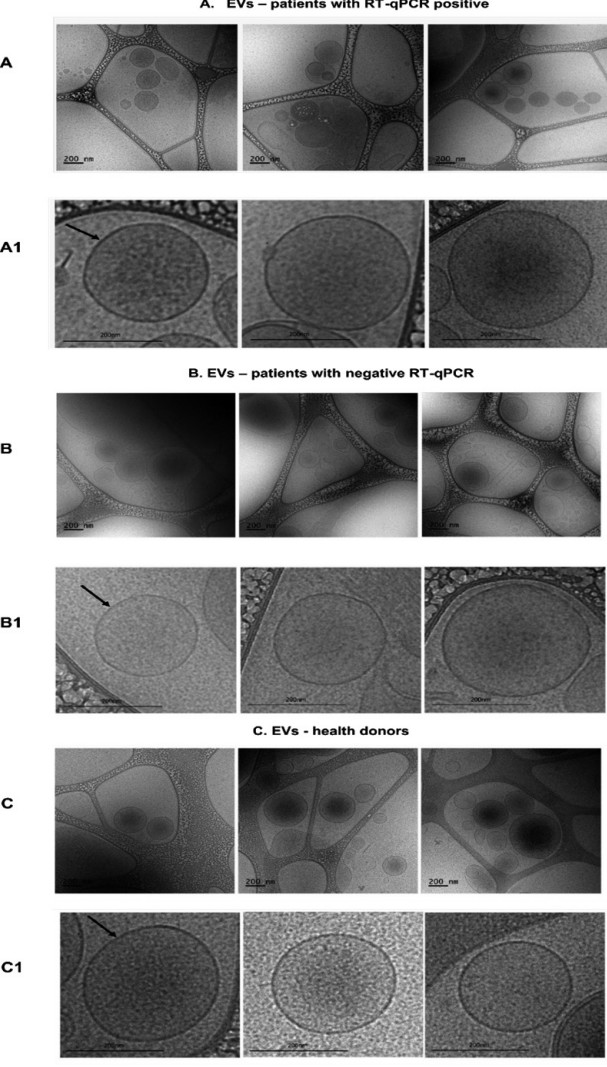

## SECOND REVIEW ROUND

REVIEWERS' COMMENTS

### REVIEWER #1

Reviewer comments: I am pleased to inform you that the manuscript has been carefully revised in accordance with the previous comments and suggestions. All concerns have been addressed, and the manuscript now fully complies with the guidelines of Memórias do Instituto Oswaldo Cruz (MIOC).

