## [Reviewer Report · FIRST REVIEW ROUND - REVIEWERS COMMENTS]

## REVIEWER #1

Reviewer comments: The manuscript presents a well-designed study investigating the role of extracellular vesicles (EVs) in COVID-19 and sepsis, with a focus on their characterization, viral cargo, and clinical correlations. The study is timely and addresses an important gap in understanding the pathophysiology of severe COVID-19. The methodologies are robust, combining nanoparticle tracking analysis (NTA), flow cytometry, RT-qPCR, and cryo-EM. However, some aspects require clarification, and additional references would strengthen the discussion. Below are specific comments organized by section. 

## a) Abstract Adequacy

The abstract provides a concise overview of the study’s objectives and key findings. It clearly states the investigation of EV characteristics in COVID-19 and sepsis patients, including their size, concentration, and viral cargo. However, it could be strengthened by briefly mentioning the clinical implications of the findings, particularly the association between CD81+ EVs and patient outcomes. Adding a sentence about the potential diagnostic or prognostic value of these EVs would enhance its impact.

## b) Originality and Importance

This study makes several significant contributions to the field. First, it provides novel evidence that EVs from COVID-19 patients can carry SARS-CoV-2 RNA, including the Gamma variant, which has not been extensively reported. Second, the identification of CD81+ EVs as a potential biomarker for mortality is clinically relevant and could inform future prognostic tools. The study also advances technical methodologies by combining NTA, flow cytometry, and cryo-EM for comprehensive EV characterization. These findings are particularly important given the ongoing need to understand the pathophysiology of severe COVID-19 and sepsis, as well as the role of EVs in these conditions. 

## c) Methodology, Results, and Discussion

The methodology is robust, employing multiple techniques to characterize EVs, including NTA for size and concentration, flow cytometry for cellular origin, and RT-qPCR for viral RNA detection. The use of cryo-EM to visualize EVs adds valuable morphological data. However, the study could benefit from additional details on EV isolation protocols, such as steps taken to minimize lipoprotein contamination, and clarification on whether isotype controls were used in flow cytometry experiments to ensure antibody specificity. 

The results are well-presented and demonstrate clear differences in EV profiles between patients and controls. The detection of viral RNA in EVs, particularly the nucleocapsid gene, is a notable finding. However, the discussion could be expanded to explore potential mechanisms by which EV-associated viral RNA might contribute to disease progression, such as through immune activation or endothelial dysfunction. Comparing these findings to existing literature on EVs in bacterial sepsis would also provide broader context.

## d) References

The reference list is generally appropriate but could be strengthened by including additional key studies. For example, recent work on EV-mediated endothelial injury in COVID-19 (DOI: 10.1002/jev2.12117) and the role of EVs in sepsis-associated ARDS (DOI: 10.3389/fimmu.2023.1150564) would provide valuable context. Additionally, citing the MISEV guidelines (DOI: 10.1002/jev2.12404) would underscore the study’s adherence to current EV research standards.

## e) Figures and Tables

The figures and tables are clear and effectively support the study’s findings. Figure 1 provides a good visualization of EV size and concentration differences between groups, while Figure 5’s ROC curve effectively highlights the prognostic value of CD81+ EVs. However, annotating the cryo-EM images to highlight specific features, such as double-membrane structures or putative viral particles, would enhance their interpretability. Table III is comprehensive but could include interquartile ranges for laboratory values to better represent data variability.

## Overall Strengths

- Comprehensive multi-method approach to EV characterization.

- Novel findings on EV-associated viral RNA and its clinical implications.

- Strong statistical analysis, including ROC curves for prognostic evaluation.

## Areas for Improvement

- Provide more details on EV isolation and purity controls.

- Expand the discussion to include mechanistic hypotheses and comparisons with bacterial sepsis.

- Update references to include recent studies on EVs in COVID-19 and sepsis.

## Recommendation

This manuscript presents valuable findings that advance our understanding of EVs in COVID-19 and sepsis. With minor revisions to address the points above, it would be suitable for publication.

## AUTHORS RESPONSE TO THE REVIEWERS

Reviewer 1

Reviewer’s comments: The manuscript presents a well-designed study investigating the role of extracellular vesicles (EVs) in COVID-19 and sepsis, with a focus on their characterization, viral cargo, and clinical correlations. The study is timely and addresses an important gap in understanding the pathophysiology of severe COVID-19. The methodologies are robust, combining nanoparticle tracking analysis (NTA), flow cytometry, RT-qPCR, and cryo-EM. However, some aspects require clarification, and additional references would strengthen the discussion. Below are specific comments organized by section.

ANSWER: We would like to thank you for the critical analysis of our manuscript. We added all suggestions to improve the manuscript for publication. The corrections on the manuscript were in highlight in yellow.

## a. Abstract Adequacy

Reviewer’s comments:

The abstract provides a concise overview of the study’s objectives and key findings. It clearly states the investigation of EV characteristics in COVID-19 and sepsis patients, including their size, concentration, and viral cargo. However, it could be strengthened by briefly mentioning the clinical implications of the findings, particularly the association between CD81+ EVs and patient outcomes. Adding a sentence about the potential diagnostic or prognostic value of these EVs would enhance its impact.

ANSWER: We do agree with the referee. We aimed to include this information, while keeping the abstract restrict to 200 words, as requested by the MIOC.

Lines 42 to 51- “RESULTS: NTA showed an increased concentration of microparticles in patients. RT-qPCR analysis of EVs detected the virus in 14 samples, two of which were consistent with the Gamma variant. Extracellular vesicles (EVs) predominantly derived from T cells and platelets, with increased CD81 expression in individuals who died. Cryo-EM revealed extracellular vesicles with an average size of 200 nm. CONCLUSIONS: Our findings suggest that patients’ extracellular vesicles likely harbour viral components, suggesting their potential role as carriers of SARS-CoV-2, and that extracellular vesicles from deceased patients demonstrate elevated CD81 expression.”

## Originality and Importance

Reviewer’s comments:

This study makes several significant contributions to the field. First, it provides novel evidence that EVs from COVID-19 patients can carry SARS-CoV-2 RNA, including the Gamma variant, which has not been extensively reported. Second, the identification of CD81+ EVs as a potential biomarker for mortality is clinically relevant and could inform future prognostic tools. The study also advances technical methodologies by combining NTA, flow cytometry, and cryo-EM for comprehensive EV characterization. These findings are particularly important given the ongoing need to understand the pathophysiology of severe COVID-19 and sepsis, as well as the role of EVs in these conditions.

ANSWER: We thank the referee for the comment.

## Methodology, Results, and Discussion

Reviewer’s comments:

The methodology is robust, employing multiple techniques to characterize EVs, including NTA for size and concentration, flow cytometry for cellular origin, and RT-qPCR for viral RNA detection. The use of cryo-EM to visualize EVs adds valuable morphological data. However, the study could benefit from additional details on EV isolation protocols, such as steps taken to minimize lipoprotein contamination, and clarification on whether isotype controls were used in flow cytometry experiments to ensure antibody specificity.

ANSWER. We described the protocol “Purification of specific EVs by size exclusion chromatography (SEC)” in the Lines 222 to 230. The Sepharose CL-4B column increased the size exclusion separation of extracellular vesicles (EVs), efficiently removing soluble proteins and smaller lipoproteins, including LDL (20–30 nm), VLDL (30–80 nm), and HDL (7–12 nm). Consequently, samples enriched in extracellular vesicles were acquired, exhibiting most particles in size ranges above those of these lipoproteins. Subsequent analysis via NTA and cryo-electron microscopy verified that nearly 95% of the isolated particles were above the diameters associated with these contaminants. 

## Results section

We add Cryo-Electron Microscopy (Cryo-EM). We used cryo-electron microscopy (Cryo-EM), which allows for the visualization of EVs while preserving their membranes in a near-native state, revealing the morphology and size of the EVs. We obtained images of EVs from samples that tested positive and negative in RT-qPCR for SARS-CoV-2 genes, as well as EVs from healthy donors, showing morphological similarities within each group. Most EVs had a rounded shape (Fig. 7 A, B, and C). The complete images (Fig. 7 A1, B2, and C2), approximately 200 nm in size, show a distinct outer layer, the lipid bilayer characteristic of extracellular vesicles (29,33); inside, the internal phase of the vesicles is evident, potentially which includes intracellular materials such as proteins, RNA, or other biomolecules. Unfortunately, we are not to determine or visualize the presence of SARS-CoV-2 in the images 

## Discussion

Our EV analyses via Cryo-EM revealed similar results, showing a predominance of EVs with well-defined lipid bilayers in the membranes. Most EVs had a rounded shape and an average size of 200 nm. This finding aligns with previous reports. Emelyanov et al. (2020), evaluating EV morphology in cerebrospinal fluid samples via cryo-electron microscopy (Cryo-EM), found that more than 80% of EVs had a lipid bilayer/membrane and a rounded shape. The remaining EVs had an elongated shape (28). Previously, in 2014, Arraud et al., when analyzing plasma samples from healthy donors with isolated EVs, obtained Cryo-EM images showing EVs with a well-defined outer layer in two lines, round-shaped EVs (29).

(comments): The results are well-presented and demonstrate clear differences in EV profiles between patients and controls. The detection of viral RNA in EVs, particularly the nucleocapsid gene, is a notable finding. However, the discussion could be expanded to explore potential mechanisms by which EV-associated viral RNA might contribute to disease progression, such as through immune activation or endothelial dysfunction. Comparing these findings to existing literature on EVs in bacterial sepsis would also provide broader context.

ANSWER. We added in the discussion. “Extracellular vesicles (EVs) are biological structures that carry a wide range of molecules, including mRNAs, microRNAs, and other non-coding RNAs, which play crucial roles in the progression and modulation of various pathological processes. Recent evidence indicates that the molecular composition of EVs can vary according to the pathophysiological context, thereby reflecting the state of the parental cell and influencing disease evolution (41; 42; 43; 44). Particular attention has been devoted to EVs transporting viral RNA, as these vesicles can activate the immune system, modulating gene expression, interfering with cellular functions, and facilitating infectious processes, thereby promoting intercellular communication across tissues and distances (45; 46). Such mechanisms of EV-mediated RNA transfer are being extensively investigated not only to elucidate the potential of vesicle-associated viral RNA to initiate new replication cycles (47), but also to better understand their involvement in the clinical course of viral infections. Moreover, the identification of specific EV-associated biomarkers emerges as a promising strategy for the development of innovative diagnostic approaches (48; 49). Regarding microRNAs, EVs containing specific miRNA cargo also rely on these communication mechanisms to exert relevant regulatory functions. Notably, these vesicles have been reported to reduce mortality in certain pathological contexts (50) and to confer cardioprotective effects in animal models of sepsis (51), highlighting their therapeutic potential.” 

## d) References

The reference list is generally appropriate but could be strengthened by including additional key studies. For example, recent work on EV-mediated endothelial injury in COVID-19 (DOI: 10.1002/jev2.12117) and the role of EVs in sepsis-associated ARDS (DOI: 10.3389/fimmu.2023.1150564) would provide valuable context. Additionally, citing the MISEV guidelines (DOI: 10.1002/jev2.12404) would underscore the study’s adherence to current EV research standards.

ANSWER In this study, we employed standardized methodologies for extracellular vesicle (EV) isolation and characterization. Specifically, ultracentrifugation and column-based particle separation were performed, following the guidelines established by the Minimal Information for Studies of Extracellular Vesicles (MISEV 2023, reference number 32. Welsh J, Goberdhan DC, O’Driscoll L, Buzas EI, Blenkiron C, Bussolati B, et al. Minimal information for studies of extracellular vesicles (MISEV2023): from basic to advanced approaches. J Extracell Vesicles. 2024;13(2):e12404. doi:10.1002/jev2.12404 in the Manuscript’s references). These approaches were selected to ensure methodological rigor, reproducibility, and compliance with current international recommendations. Furthermore, the inclusion of these procedures and their detailed citation were carried out in accordance with the reviewer’s suggestions, thereby strengthening the methodological robustness and alignment with the most up-to-date consensus in the field.

## e) Figures and Tables

The figures and tables are clear and effectively support the study’s findings. Figure 1 provides a good visualization of EV size and concentration differences between groups, while Figure 5’s ROC curve effectively highlights the prognostic value of CD81+ EVs. However, annotating the cryo-EM images to highlight specific features, such as double-membrane structures or putative viral particles, would enhance their interpretability. Table III is comprehensive but could include interquartile ranges for laboratory values to better represent data variability.

ANSWER. We added arrows to the Cryo-EM images to indicate the lipid bilayer of the EVs in the samples from patients with positive and negative RTq PCR and healthy donors.

Protocol -Cryo-Electron Microscopy (Cryo-EM). We used cryo-electron microscopy (Cryo-EM), which allows for the visualization of EVs while preserving their membranes in a near-native state, revealing the morphology and size of the EVs. We obtained images of EVs from samples that tested positive and negative in RT-qPCR for SARS-CoV-2 genes, as well as EVs from healthy donors, showing morphological similarities within each group. Most EVs had a rounded shape (Fig. 7 A, B, and C). The complete images ([Fig f8]), approximately 200 nm in size, show a distinct outer layer, the lipid bilayer characteristic of extracellular vesicles (29,33); inside, the internal phase of the vesicles is evident, potentially which includes intracellular materials such as proteins, RNA, or other biomolecules. Unfortunately, we are not to determine or visualize the presence of SARS-CoV-2 in the images

---

## [Reviewer Report · REVIEWERS COMMENTS]

## REVIEWER #1

Reviewer comments: I am pleased to inform you that the manuscript has been carefully revised in accordance with the previous comments and suggestions. All concerns have been addressed, and the manuscript now fully complies with the guidelines of Memórias do Instituto Oswaldo Cruz (MIOC).